# Nutritional Management of Idiopathic Nephrotic Syndrome in Pediatric Age

**DOI:** 10.3390/medsci11030047

**Published:** 2023-07-28

**Authors:** Graziana Lella, Luca Pecoraro, Elisa Benetti, Olivia Chapin Arnone, Giorgio Piacentini, Milena Brugnara, Angelo Pietrobelli

**Affiliations:** 1Pediatric Clinic, Department of Surgical Sciences, Dentistry, Gynecology and Pediatrics, University of Verona, 37126 Verona, Italyoliviarnone@gmail.com (O.C.A.); angelo.pietrobelli@univr.it (A.P.); 2Pediatric Nephrology, Department of Women’s and Children’s Health, University of Padua, 35122 Padua, Italy; 3Pennington Biomedical Research Center, Baton Rouge, LA 70808, USA

**Keywords:** nephrotic syndrome, nutrition, pediatrics, dietary suggestions

## Abstract

Nephrotic syndrome (NS) is a common pediatric disease characterized by a dysfunction in the glomerular filtration barrier that leads to protein, fluid, and nutrient loss in urine. Corticosteroid therapy is the conventional treatment in children. Long-term complications of NS and prolonged exposure to steroids affect bones, growth, and the cardiovascular system. Diet can play an important role in preventing these complications, but there is a scarcity of scientific literature about nutritional recommendations for children with NS. They need individualized nutrition choices not only during the acute phase of the disease but also during remission to prevent the progression of kidney damage. The correct management of diet in these children requires a multidisciplinary approach that involves family pediatricians, pediatric nephrologists, dietitians, and parents.

## 1. Introduction

Nephrotic syndrome (NS) in children is characterized by proteinuria (≥40 mg/m^2^/h or ≥300 mg/dL or 3+ on a urine dipstick or urine protein–creatinine ratio ≥2000 mg/g or ≥200 mg/mmol), hypoalbuminemia, edema, and hyperlipidemia [1]. The most frequent form in childhood is idiopathic NS, which usually develops after the first year of life, with an incidence of 2–7 per 100,000 and a prevalence of nearly 16 per 100,000 worldwide [2]. The most common histologic pattern associated with idiopathic NS is minimal change disease (MCD), which accounts for 70–90% of patients [3]; however, the incidence of focal segmental glomerulosclerosis (FSGS) is increasing [4]. More recently, MCD and FSGS have been considered on a spectrum of a single disease process, in which MCD is an earlier and more responsive phase, and FSGS represents a more advanced and resistant stage [5]. Regardless of the underlying etiology, the structural changes and the subsequent dysfunction of the glomerular filtration barrier caused by these abnormalities lead to protein, fluid, and nutrient loss in urine, which is responsible for signs and symptoms of NS [5]. The clinical classification of idiopathic NS is based on response to corticosteroid therapy, which is the conventional treatment in children. A total of 85% of patients achieve complete remission [6], as MCD is responsive to steroids in 93% of cases, and other histological patterns may also respond in 25–50% of cases [7]. As a result, steroid therapy is initiated without histological confirmation by kidney biopsy. However, 70–80% of these children show at least one relapse during the follow-up, and 50% experience frequent relapses, becoming steroid-dependent. The long-term prognosis is good, as NS resolves spontaneously following puberty. In 10–30% of cases, it may progress to adulthood, while continuing to respond well to corticosteroids [6]. Only a small percentage of children, approximately 10%, may become secondarily steroid-resistant in subsequent relapses and progress towards kidney failure [8]. The major prognostic factors are the patient’s response to treatment and the frequency of the relapses; therefore, the goal of the management of NS is to achieve the complete absence of proteinuria and to preserve kidney function [1]. As the disease can have a chronic relapsing–remitting course, its complications arise mainly from the toxic effects of long-term or frequent use of steroids. As a result, the effective management of relapses represents a great challenge. NS typically presents with peripheral, gravity-dependent edema, or anasarca in severe cases, which can lead to acute complications including ascites, pulmonary edema, or pleural effusion. The reduced oncotic pressure causes intravascular volume depletion, and acute kidney injury may also occur. Acute complications related to the nephrotic state include infections and hypercoagulability, with an increased risk of thromboembolic disease [5]. Long-term sequelae of NS and prolonged exposure to corticosteroids affect bones, growth, and the cardiovascular system, with resulting hyperlipidemia, hypertension, hypercoagulability, anemia, vitamin D deficiency, and secondary hyperparathyroidism [2]. Therefore, diet can play an important role in narrowing and preventing complications of NS: making dietary changes is crucial in replacing nutrient losses and correcting metabolic abnormalities, but also in avoiding renal disease progression [9,10]. There is a scarcity of scientific literature regarding nutritional recommendations for children with NS. Standardized diet recommendations are needed to manage the condition and the side effects of steroid treatment [11]. The aim of this article is to deepen the state of the art knowledge of nutritional management of these patients, who present unique nutrition support needs and require individualized nutritional choices, not only during the acute phase of the disease but also during the periods of remission, to prevent the progression of kidney damage.

## 2. Acute Phase/Relapse of NS

### 2.1. Fluid Balance

Patients with NS experience fluid retention and edema, and this leads to an overall water imbalance in the body [9]. Nevertheless, fluid restriction for edema is usually not recommended, as it may cause hypotension and acute kidney injury (AKI), worsening intravascular volume depletion and dehydration [1,12]. However, moderate fluid restriction can be advised with caution in selected cases, such as in patients who develop significant hyponatremia, massive anasarca, or oliguric renal failure [12,13,14]. The management of edema in NS first requires the assessment of the euvolemic state of the patient. In the case of normal intravascular volume, moderate edema should be treated only with a low-salt diet, without fluid restriction. Severe edema requires fluid restriction with loop diuretics in hospital settings. In case of reduced intravascular volume with normal blood pressure, albumin should be administered intravenously, followed by furosemide once euvolemia is restored. Hypovolemic shock should be treated following specific resuscitation guidelines [6]. All foods that are liquid at room temperature, such as milk, juice, yogurt, ice cream, soup etc., should be counted upon evaluation of fluid intake [9]. A few strategies can be implemented to control fluid intake in children, such as using small glasses filled to look like they contain a greater amount of fluid, avoiding salty foods that increase thirst, offering frozen pieces of fruit or chewing gum to quench thirst, and avoiding warm environment.

### 2.2. Macronutrients Intake

#### 2.2.1. Carbohydrates

Although corticosteroids are the cornerstone of treatment for NS, their prolonged and repeated use may lead to significant adverse effects, such as hyperglycemia and insulin resistance [13]. Corticosteroids can also cause weight gain, and subsequently obesity, due to behavioral changes, including increased appetite [15,16]. Obesity affects patients’ quality of life and could seriously impact emotional health and social relationships in the future as adults [16]. Because of this, the short- and long-term effects of steroid therapy on body weight must be discussed with patients and their families [15]. Children and their parents need to be instructed to follow a healthier diet [14], with a focus on a reduced intake of simple sugars [11], while an adequate intake of high-complex carbohydrates should be ensured to maximize the utilization of proteins [7,10].

#### 2.2.2. Proteins

NS causes protein loss through the damaged glomerular filtration barrier in the urine. Early management of the NS recommended an increased protein intake to replace losses and avoid the development of protein malnutrition [13]. However, recent studies demonstrate that increased dietary protein intake does not improve serum albumin concentrations [13]. The higher dietary protein intake results in increased urinary protein losses without a net gain of protein, due to the altered glomerular permselectivity. In addition, a high-protein diet leads to changes in glomerular hemodynamics that may accelerate the progression of renal disease [10]. On the contrary, protein restriction can positively impact kidney function in adult patients with decreased renal function, but a very low-protein diet should be avoided for the risk of malnutrition [1]. Intake of high-quality proteins is recommended in patients with proteinuria, as it is recommended for the general pediatric population [6,14]. Vegetable sources of protein are preferred whenever possible [1].

#### 2.2.3. Lipids

Dyslipidemia is a frequent metabolic complication in patients with active NS. It is caused by compensatory protein and lipoprotein synthesis in the liver in response to urinary protein loss, reduced transport of cholesterol in the bloodstream due to hypoalbuminemia, and an acquired deficiency of enzymes involved in the regulation of lipid metabolism, which are lost in urine. Additionally, corticosteroid use may be associated with an elevation in blood lipid levels [13]. The long-term effects of dyslipidemia in pediatric NS are unclear. To date, no data support a link between dyslipidemia in these patients and an increased incidence of cardiovascular disease in adulthood. However, the acceleration of the atherosclerotic process in pediatric NS probably has multifactorial roots, as these children show other atherogenic risk factors, such as hypoalbuminemia, hypertension, hypercoagulability, and obesity [13]. On the other hand, dyslipidemia is involved in the progression of renal disease [10]. Measuring baseline lipid levels in children with NS may be useful in screening for underlying secondary causes of dyslipidemia [17]. Lipids normalize following remission, so reducing proteinuria is usually sufficient to reduce hyperlipidemia [10]. Managing dyslipidemia in pediatric NS during the acute phase requires dietary optimization. Children over 2 years old should follow the Cardiovascular Health Integrated Lifestyle Diet (CHILD-1): fats should be restricted to <30% of total daily calories, saturated fats to <10%, and cholesterol consumption to <300 mg/d [1,13,18], while simultaneously increasing the consumption of healthier fats, such as monounsaturated, polyunsaturated, and omega-3 fatty acids [13]. On the other hand, children who also present with hyperlipidemia should follow the CHILD-2 diet plan, which further limits the intake of saturated fats to <7% and cholesterol to <200 mg/d [13]. No fat intake restriction is recommended for children under the age of 2, and they can be breastfed [13]. The use of pharmacological agents to treat dyslipidemia is controversial when dietary adjustments are insufficient [10]. Evidence of benefits and safety in the use of statin therapy to reduce serum cholesterol levels in children and adolescents with NS is lacking; therefore, it is not recommended for all patients. However, it can be considered in high-risk patients with severe low-density lipoprotein (LDL) cholesterol elevation based on clinical circumstances. Specific triglyceride (TG)-lowering agents are also not recommended for children and adolescents with elevated TGs [13]. New strategies, which may induce partial or complete clinical remission of NS, are increasingly being implemented. These include bile acid sequestrants, fibrates, nicotinic acid, ezetimibe, and lipid apheresis. Because there is a dysregulation in some proteins involved in the lipid metabolism in NS, such as proprotein convertase subtilisin/kexin type 9 (PCSK9), the use of target therapy with anti-PCSK9 monoclonal antibodies or small inhibitory ribonucleic acids (RNAs) could play a crucial role for future treatments [19]. Studies concerning dietary supplements, such as fish oils and policosanol, in patients with NS are limited. Following omega-3 fatty acid supplementation in patients with NS, the literature shows a decrease in postprandial chylomicrons, a small decrease in serum triglycerides, and mixed effects on LDL levels [19].

### 2.3. Micronutrients Intake

#### 2.3.1. Sodium

Sodium plays a key role in regulating blood pressure and fluid retention in patients with NS. Nevertheless, there is a lack of standardized recommendations for sodium intake in children with newly diagnosed NS. In children with NS, current suggestions for sodium restriction vary from <2 mEq/kg/d to an approach based on a “no added salt diet” [6,11]. The Pediatric Nephrology Clinical Pathway Development Team proposes a one-to-one ratio of 1 mg of sodium for each calorie (kcal) in order to adequately restrict sodium intake to energy requirement [11]. During the initial nutrition consultation, emphasis should be placed on strategies to lower sodium in the diet, preferring fresh foods to processed ones [11], identifying and limiting high-sodium foods, and avoiding salt when preparing food or eating [9].

#### 2.3.2. Calcium and Vitamin D

Metabolic bone disease (MBD) is a frequent complication of NS in children. The pathogenesis is multifactorial. Urinary loss of minerals and plasma proteins, including calcium and vitamin D binding protein, results in hypocalcemia and low vitamin D levels, which may lead to osteopenia and osteoporosis. Corticosteroids decrease intestinal absorption and tubular reabsorption of calcium. Hypocalcemia is usually not long-lasting and serum levels of calcium can normalize during remission, but prolonged corticosteroid therapy, especially due to frequent relapses, may cause MBD. Corticosteroids also suppress the development and function of osteoblasts, as they increase the lifetime of osteoclasts and inhibit the release of parathyroid hormone, which results in a reduced overall bone mineral density [13]. Serum vitamin D levels should be routinely monitored in children with NS starting at the time of diagnosis [20]. Periodic assessments are also indicated for serum phosphorus, ionized calcium, parathyroid hormone, and alkaline phosphatase [13]. Moreover, dual-energy X-ray absorptiometry (DXA) can be considered in patients with NS to measure bone mineral density [14]. Patients and their caregivers should be counseled to monitor calcium and vitamin D intake in order to have an age-appropriate calcium intake [11]. When dietary modification is not successful, patient-specific calcium and vitamin D supplementation should be prescribed. Daily supplementation with 500 mg of elemental calcium (250 mg twice daily) is advised [11,13]. Addressed hypocalcemia, vitamin D supplementation regimens reported in the literature range from 800–1000 IU of cholecalciferol daily to 60,000 IU once a week [11,13]. In patients with advanced renal insufficiency, 1,25-dihydroxycholecalciferol should be used for vitamin D replacement [10]. There is insufficient evidence on the pharmacologic treatment of MBD in children with NS. The prevention and treatment of steroid-induced osteoporosis in pediatric age are currently based on the reduction or discontinuation of steroids [14].

#### 2.3.3. Iron, Copper, and Zinc Deficiency and Anemia

The urinary loss of transferrin, erythropoietin, transcobalamin, ceruloplasmin, iron, and trace elements may lead to anemia. Patients with iron deficiency anemia should receive replacement therapy, and, in the case of low erythropoietin levels, therapy with erythropoietin should be considered. Transferrin levels will correct with the resolution of proteinuria [13]. Laboratory evidence of anemia that does not respond to iron or erythropoietin therapy suggests deficiencies in other micronutrients, like copper, zinc, and vitamin B12. Copper and zinc deficiencies result in reduced activity of copper and zinc superoxide dismutase, shortening the life span of red blood cells. Moreover, the addition of zinc therapy to the standard therapy of NS seems to reduce the number and the frequency of relapses, to help induce remission [13], and to reduce the proportion of infections associated with relapses, with a metallic taste as a mild adverse event [21]. The mechanism of zinc action is not fully clear, but it is probably linked to its immunoregulatory role: zinc deficiency might lead to the down-regulation of Th1 cytokines, with an increased risk of infections [21]. Table 1 provides a summary of dietary recommendations to follow during the acute phase of NS in children, based on the literature examined.

## 3. NS in Complete Remission

### 3.1. Fluid Balance and Macronutrients Intake

Once edema has resolved, no fluid restriction is needed [13]. The chronic, relapsing course of the disease leads to repeated protein loss. Therefore, NS may affect growth and development in children. In addition, in children with frequent relapses who are exposed to frequent use of steroids, growth may be impacted negatively. The result could be a reduced weight or height standard deviation score (SDS) [13]. Diet should guarantee daily protein intake recommended for the general pediatric population, even during remission [6,14]. Dyslipidemia does not usually require medication since it improves with the resolution of proteinuria and discontinuation of corticosteroids [18]. However, additional studies have shown that the hyperlipidemic profile may persist in remission. Therefore, the restriction of saturated and trans-fat is recommended during remission [13]. Moreover, fatty acids (FAs) play an important biological role in cell-to-cell communication during inflammatory processes, including in NS, given the immunological pathogenesis of this disease. When analyzing the FA profile in children with NS in stable remission, persistently higher blood levels of linoleic and arachidonic acid can be found; therefore, these FAs could be considered as biomarkers of the risk of relapse in children with NS [22].

### 3.2. Micronutrients Intake

In remission, children can follow a less-restrictive “maintenance diet,” in which avoidance of added salt and snacks containing high levels of salt is recommended, but the consumption of sodium-containing foods is allowed in small amounts [18]. This dietary regimen prevents iatrogenic hypertension and kidney damage, as well as weaning the habit of salty taste, since salt must be removed during relapses. The reported prevalence of hypertension in pediatric steroid sensitive SN (SSNS) ranges from 7% to 34%, especially in children with steroid dependent NS (SDNS) and frequently relapsing NS (FRNS), during remission. The multifactorial etiology is linked to prolonged corticosteroid use and fluid imbalance; however, a positive familial history can usually be recognized. Children with chronic hypertension should be treated according to current hypertension guidelines [6]. Hypocalcemia usually normalizes following remission, so, in this phase, ensuring adequate dietary calcium intake is usually sufficient [13]. However, long-term oral calcium supplementation (250–500 mg daily) could be necessary in the case of prolonged corticosteroid therapy or insufficient calcium intake [6,18]. Assessment of 25-OH-vitamin D annually in patients with SDNS or FRNS (after three months of remission) is recommended, maintaining levels >20 ng/mL (>50 nmol/L). If vitamin D deficiency is detected, it should be treated [6].

## 4. Special Diets

Recently, many attempts have been made to find alternative dietary regimens that may be helpful in treating this disease. Successful treatment of NS with hypoallergenic diets is reported in the literature, but their use is currently limited. Although the mechanism is not fully clear, elimination diets may modulate the composition and immune function of gut microbiota, lowering the risk of certain immune-mediated diseases, such as NS [23]. Foods responsible for sensitivities include milk, chicken, egg white, gluten, pork, wheat flour, and beef. Trials with specific elimination diets based on intradermal testing with various food allergens in patients with NS showed encouraging results, leading to the remission of NS [23]. Recently, interest in the association between food allergy/sensitivity and NS has been re-examined, with a focus on gluten-free and dairy-free diets. A gluten-free diet may stabilize the podocyte cytoskeleton and modify the gut microbiota, reducing the release of inflammatory mediators responsible for increased glomerular permeability to protein, which results in reduced protein urinary loss [13]. A soy-based vegetarian diet leads to decreased serum levels of cholesterol and apolipoprotein and reduced proteinuria due to lower protein and lipid intake [19]. Moreover, a soy-protein diet improves the oxidative status of NS, thanks to its isoflavones, which reduce glomerular damage [24]. Similarly, the vegan diet has been shown to significantly reduce urinary protein loss due to the limited protein intake and its vegetable origin. Moreover, a vegan diet improves the hyperlipidemic state typical of NS, preventing the progression of kidney damage [25]. However, the vegan diet requires supplementation of specific nutrients to avoid malnutrition, so a pediatric dietitian should be consulted before undertaking a plant-based diet. Patients with NS may benefit from a Mediterranean diet. As previously mentioned, the FA profiles of patients with NS show an increase in the pro-inflammatory omega-6 series, even during remission. In these patients, adherence to the Mediterranean diet modifies the omega-6/omega-3 ratio in favor of the anti-inflammatory omega-3 and decreases blood linoleic acid levels [26]. In Persian medicine, quince, pomegranate, and whole grains could be recommended for NS. According to several studies, pomegranate may have nephroprotective effects, stabilizing kidney function and reducing protein excretion. Moreover, pomegranate has antioxidant and anti-inflammatory properties. Quince is rich in flavonoids and antioxidants, which help improve glomerular dysfunction, decrease urine protein excretion, increase nitric oxide levels, and reduce the concentrations of renin and angiotensin, lowering systolic blood pressure. Studies have shown that whole grain consumption could positively affect kidney function thanks to its beneficial effects on the microbiome. NS is an immunological disorder, and dysbiosis of gut microbiota contributes to immunological disorders [27]. Daily use of whole wheat promotes the growth of intestinal microbiota, which may have a protective effect on the kidney either directly or indirectly by influencing the development and progression of CKD. The previously mentioned studies suggest that pomegranate, quince, and barley compounds can be added to the diet of children with NS [27,28].

## 5. Conclusions

NS is a complex disease requiring special attention to nutrition through a multidisciplinary approach that involves several professional figures, such as dietitians, pediatric nephrologists, family pediatricians, and parents. Specifically, the dietitian with expertise in pediatric nephrology has a fundamental role and supports the pediatric nephrologist in developing a personalized diet with medical decisions, practices, interventions, and products tailored to the individual patient. A regular assessment of dietary intake by the dietitian throughout the course of NS should be guaranteed. Family pediatricians should also collaborate closely with parents, ensuring close follow-up during remission and prompt recognition of relapses to implement all the necessary strategies for their management. Parents play an essential role in the healthcare of their children; therefore, clear instructions and information on diet management should be provided both during the acute and remission phases. The family-centered approach should also consider the needs of children, as regimens that are too restrictive lead to negative consequences from both a physical and psychological point of view. This “team approach” with the family as the center for the promotion of the child’s nutritional health is fundamental for successful care.

## Figures and Tables

**Table 1 medsci-11-00047-t001:** Summary of nutrition modifications suggested in the treatment of the acute phase of NS in children.

**Fluids**	In patients with maintained intravascular volume: moderate edema requires no fluid restriction; severe edema requires fluid restriction and loop diuretics in hospital settings [6]. In patients with contracted intravascular volume with normal blood pressure: administer albumin infusion, followed by furosemide [6]. In patients with hypovolemia: follow specific resuscitation guidelines [6].
**Carbohydrates**	Reduced intake of simple sugars [11], with an adequate intake of high-complex carbohydrates [7,10].
**Protein**	Daily protein intake recommended for the general pediatric population [6,14]. Prefer vegetable sources.
**Dietary fat**	In children >2 years old: <30% of total calories, saturated fats <7–10%, cholesterol consumption <200–300 mg/d [1,13,18], increasing the consumption of monounsaturated, polyunsaturated, and omega-3 fatty acids [13]. In children <2 years old: no fat intake restriction [13]. Children with hyperlipidemia: <7% saturated fats, <200 mg/d cholesterol [13].
**Sodium**	<2 mEq/kg/d [11]. “No added salt diet” approach [11]. 1 mg for each kcal [11].
**Calcium and vitamin D**	Elemental calcium: 500 mg daily (250 mg twice daily) [11,13]. Cholecalciferol: from 800–1000 IU daily to 60,000 IU once a week [11,13]. In patients with advanced renal insufficiency, use 1,25-dihydroxycholecalciferol [10].
**Iron, copper, and zinc**	In patients with iron deficiency anemia: administer replacement therapy [13]. In patients with low erythropoietin levels: consider therapy with erythropoietin [13]. In patients with anemia that does not respond to iron and erythropoietin therapy: consider and correct deficiencies in other micronutrients, like copper, zinc, and vitamin B12 [13].

## Data Availability

Not applicable.

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
