# Peer review of "Nutritional Management of Idiopathic Nephrotic Syndrome in Pediatric Age"

_medsci, 2023, doi:10.3390/medsci11030047_

Round 1

Reviewer 1 Report

Dear Authors, thank you for the opportunity to review your paper.  While I found your submission a topic of interest and clinical relevance, I did find that the overall structure (along with language and sentence structure) very confusing.  There is information that is taken out of context  for example:  energy information meant for adults (kcal/kg) from KDIGO which was applied to pediatric population inappropriately.  Many children in the pediatric stage would not be well served with caloric intake at 35 kcal/kg.  Furthermore, guidance is provided for increasing protein intake in the case of ongoing proteinuria.  This guidance appears to be adult-based in KDIGO and not typical in the pediatric setting/nor specified in the KDIGO chapter on pediatrics.   

While the topic of your article is nutrition for NS in pediatrics there is some blurring of nutrition recommendations for CKD in pediatrics.  While CKD may be the long-term consequence of NS in pediatrics, the manner in which you speak to the CKD is very misleading.  Perhaps referring the reader to explore nutrition for CKD when the child with NS progresses to this stage would be more appropriate than adding CKD to this article.

As a clinician in pediatric nephrology, I know and understand the nutritional management of NS in pediatrics; however, another clinician looking to use your article as a reference for may be VERY confused by its content.  There are several contradictions (for example: use of fluid restriction...yes or no).  In the paragraph under fluid, there is suggestion that fluid restriction is not necessary while later in the paragraph there is guidance provided for strategies for restricting fluids.  

The ordering of your paragraphs are also misleading.  In section 2 under Energy and Macronutrients, protein is listed and addressed ahead of Energy Intake.  Of the 3 macronutrients, you address protein and lipids (assume you are referring to dietary fat not the serum values) while there is no mention of a section for carbohydrates.  

As for the section headed as micronutrients, I typically do not refer to Sodium, Calcium or vitamin D as such.  

While your article presents many important nutrition and diet concepts relevant to pediatric NS, the language and grammar used detracts from its overall content.  Sentence structure is clunky and there are incomplete sentences in the article.  There are also sentences which use incorrect verbs.  For example, in a sentence which refers to my own research (line221) you suggest that caregiver should be instructed to "improve" calcium and vitamin D intake when in fact the goal was achieve a calcium intake equal to the recommended daily amount for age plus 500 mg.  The reader would not likely glean this from the manner in which it is written.  

Under the Special diets section, you provide an overview of some of the proposed diet modifications including the Persian Medicine recommendations.  In this section, there is a statement made that "whole grains consumption can improve CKD through its beneficial effects on the microbiome".  I would suggest that a definitive sentence such as this, that has no reference provided, is a stretch at best.  

The Tables (1 and 2) are currently presented in reverse order.  Table 2 as currently presented will leave your reader with more questions than answers.  Again, the table includes information that is relevant to the adult population which is not appropriate (not safe) for pediatrics.  Both tables would benefit from some major revisions.   Perhaps a more appropriate title for Table 1 would be Summary of Nutrition Modifications Recommended in the Treatment of NS in Children.  In Table 2, use of the heading "forbidden foods" would be more appropriately titled Foods and beverages to Avoid along with Foods to Enjoy in the second column.  

Author Response

Dear reviewer,

We are sorry that you felt the article was lacking in some areas. Anyway, we think your comments can improve the quality of the article. So, we have revised the entire article. Specifically:

- based on our aim of giving the reader a clinical approach to nutrition management in the nephrotic syndrome, we wrongly thought some areas could be simplified. An example was represented by some parts of the guidance based in KDIGO and not typical in the pediatric setting/nor specified in the KDIGO chapter on pediatrics. Another example was referred to the nutrition recommendations for CKD in pediatrics.  We revised these parts following your suggestion.

- about chapter 2, we thank the reviewer and have revised this chapter.

- we have included a reference to carbohydrate intake without including a specific section for carbohydrates, as no specific changes are required according to guidelines.

-  Sodium, Calcium and vitamin D are categorized as micronutrients. We have chosen to maintain the chapter “Micronutrients” to give the reader a 360-degree view of nephrotic syndrome nutrition.

- we have chosen to maintain the overall structure of the article, correcting the order of paragraphs. We think that a 360-degree view of the periods of remission and relapse helps the reader understand the difference in nutrition management in these two phases of nephrotic syndrome.

- according to the editor, we have improved the English language of the article through the help of a native-speaking pediatrician (listed as the new author of the article).

- Table 1 and Table 2 were revised according to the reviewer's suggestion and presented in the correct order.

I thank the reviewer for his/her dedicated time and effort in our article. The suggestions helped us revise the manuscript accordingly and improve the article's quality per se.

I hope that the revised manuscript is now suitable for publication.

Thank you for your attention.

Sincerely,

Luca Pecoraro

Reviewer 2 Report

This is a very nice paper with practical guidance for the dietary treatment of idiopathic nephrotic syndrome in children. 

My only minor comment is that some formulations are a bit harsh or blunt, and I would advise the authors to soften these a little bit:

- Page 5 line 219: "(DXA) should be performed in patients with NS". Ref 14 grades this as a C1 recommendation, so perhaps the phrase "should be" is a bit harsh. The IPNA clinical practice recommendation states that DEXA can be considered in the transition period in patients with low muscle mass, frail or low intensity fractures

- Page 6 line 250: "NS affects growth and development in children. In addition, corticosteroids also negatively impact growth in pediatric NS. The result is a reduced weight standard deviation score (SDS) and height SDS". This can be the case in children with very frequent relapses who are expose to a lot of courses of steroids, but it's not true for every single idiopathic NS patient.

- Page 6 line 269: In the maintenance diet when everything is OK, personally I wouldn't talk about "forbidden" foods, but use different wording (again a bit less harsh)

- Page 7 line 299 and further: when advising a vegeterian or vegan diet to children (especially in those with an underlying chronic or relapsing condition), I would stimulate to do this in close contact with a pediatric dietician (to avoid malnutrition of specific nutrients)

Author Response

We thank the reviewer and have revised the article following his/her suggestions.

I thank the reviewer for his/her dedicated time and effort in our article. The suggestions helped us revise the manuscript accordingly and improve the article's quality per se.

I hope that the revised manuscript is now suitable for publication.

Thank you for your attention.

Sincerely,

Luca Pecoraro

Round 2

Reviewer 1 Report

Line 44 it should read nephrotic syndrome

line 52 effective vs best

line 54  including vs like

line 59 with resulting vs including

line 64  regarding vs about

line 64 omit Despite the limited date

Lin 66-69   I would suggest that there is nothing "practical" about the information presented.  This paper repeats what is currently in the literature.  Consider including a dietitian to the author list for the "practicalities" /translation of the concepts in to practical information for the reader.

Line 72 sentence structure is poor/unclear 

Line 78  Euvolemic vs volemic

Line 84 All foods that are liquid at room temperature such as ...

Line 86  Fluid vs said

Line 89   Warm environments vs sun exposure.  Also, iced tea and lemonade are full of simple sugars not appropriate for a child taking steroids.

Line 95 Omit sentence regarding adults.  Start with thought about children who are the topic of this paper.

The overall order of this very long paragraph about energy requires revamping.  There are several sentences that speak to other concepts which are very detracting for the reader lending itself to major confusion.  

Line 106 diuresis vs dialysis

Line 135 omit For this reason, in the past.  Start sentence with Early management......

Line 144  Omit In conclusion, an adequate.  Start sentence with Intake of high-qualitty proteins...

Line 146 are vs should be

Line 147  Seems there are random thoughts about carbohydrates in the protein section.  Consider a separate section on carbohydrates especially in relation to steroids!!

Line 173  Another random sentence about Carbohydrates??

Line 175 suggestions vs proposal

Line 175 recommended food items vs healthier diet (super vague!)

Line 176  Weird sentence /thought/concept

Line 201  This paper continues to demonstrate the lack of understanding of the recommendations as they relate to pediatrics.  The pediatric age is comprised of children of many different ages/sizes.  Suggesting a sodium intake of <2 g /day is adult based.  This would be excessive for a young child.  The author have taken information NOT appropriate for pediatrics and suggested that these are recommendations.  These recommendations are not for pediatrics.  Furthermore, these recommendation are not safe in the case of small children.

Line 207 identifying vs recognizing

Table 1:

Foods to avoid should sit side by side with options that are appropriate in order for the reader to better understand the concepts (fluid, sodium, fats, simple CHO)  The table in its present form remains confusing and messy.

Line 229.  When dietary modification is not successful, patient-specific calcium and Vitamin D supplementation should be prescribed.  

Line 230  OmitAlthough there is a lack of consensus about the optimal dose.  Start sentence with Vitamin D supplementation regimens.....

Reorder this paragraph to address calcium first and then Vitamin D supplementation.

Line 236.  End sentence after NS.  These are 2 thoughts and should be separated.

Table 2  Super bulky with too much text.  Consider using a whole lot less text so that it is in fact a table.  

Also, lipids versus dietary fat.  Lipids are what we test for in the blood.  Dietary fat is what a person eats.

Line 341  If the topic is nutrition for children with NS then the very first clinician mentioned should be a dietitian who is familiar with pediatric nephrology.  There should also be a mention of regular assessment of dietary intake by the dietitian throughout the course of NS. 

Line 349 Omit This type of collaraboration.   Speak to "team approach" with the family as the centre for the promotion of the child's nutritional health that is paramount in the successful care.

English improved but some sentence structure and order within paragraphs needs work/editing.

Author Response

REPLY LETTER

Dear reviewer,

we thank you for the time and efforts his/her dedicated to our article. The suggestions helped us to revise the manuscript accordingly and also to improve the quality of the article per se.

We have revised the article following his/her suggestions. Specifically:

  • We agree that the paper is a revision of what is currently in the literature, so we have accepted the suggestion and modified the title and the section “introduction”;
  • We are sorry that you felt the paragraph about energy requirements and the table 1 were so confusing; we have decided to delete them to avoid detracting the reader with other concepts or incomplete information.
  • We accepted the suggestion to create a separate section on carbohydrates.
  • We agree on the crucial role played by the dietitian and the importance of a regular assessment of dietary intake by the dietitian throughout the course of NS, so we have remarked these concepts.
  • We have tried to reduce the text in the table and simplify it.
  • About sodium intake, we have remarked that there is a lack of standardized recommendations. We wrongly used the word “recommendation” in the previous version of the article, so we have replaced it with the word “suggestions” in the new version. In our opinion, it seems to be more appropriate to the proposals reported. Moreover, all the references cited in this section specifically refer to pediatric age.
  • We accepted and reviewed all the specific suggestions line by line.